# Association of Blast Exposure in Military Breaching with Intestinal Permeability Blood Biomarkers Associated with Leaky Gut

**DOI:** 10.3390/ijms25063549

**Published:** 2024-03-21

**Authors:** Qingkun Liu, Zhaoyu Wang, Shengnan Sun, Jeffrey Nemes, Lisa A. Brenner, Andrew Hoisington, Maciej Skotak, Christina R. LaValle, Yongchao Ge, Walter Carr, Fatemeh Haghighi

**Affiliations:** 1James J. Peters VA Medical Center, Bronx, NY 10468, USA; qingkun.liu@mssm.edu (Q.L.); zhaoyu.wang@mssm.edu (Z.W.); shengnan.sun@mssm.edu (S.S.); 2Icahn School of Medicine at Mount Sinai, New York, NY 10029, USA; yongchao.ge@mssm.edu; 3Walter Reed Army Institute of Research, Silver Spring, MD 20910, USAchristina.r.lavalle.ctr@health.mil (C.R.L.); walter.s.carr.civ@health.mil (W.C.); 4Rocky Mountain Mental Illness, Research, Education and Clinical Care, Department of Veterans Affairs, Aurora, CO 80045, USA; lisa.brenner@va.gov (L.A.B.); andrew.hoisington@va.gov (A.H.); 5Anschutz Medical Campus, University of Colorado, Aurora, CO 80045, USA; 6Department of Systems Engineering and Management, Air Force Institute of Technology, Wright-Patterson Air Force Base, OH 45433, USA

**Keywords:** intestinal permeability, leaky gut, blast, military, mTBI, brain–gut axis, microbiome

## Abstract

Injuries and subclinical effects from exposure to blasts are of significant concern in military operational settings, including tactical training, and are associated with self-reported concussion-like symptomology and physiological changes such as increased intestinal permeability (IP), which was investigated in this study. Time-series gene expression and IP biomarker data were generated from “breachers” exposed to controlled, low-level explosive blast during training. Samples from 30 male participants at pre-, post-, and follow-up blast exposure the next day were assayed via RNA-seq and ELISA. A battery of symptom data was also collected at each of these time points that acutely showed elevated symptom reporting related to headache, concentration, dizziness, and taking longer to think, dissipating ~16 h following blast exposure. Evidence for bacterial translocation into circulation following blast exposure was detected by significant stepwise increase in microbial diversity (measured via alpha-diversity *p* = 0.049). Alterations in levels of IP protein biomarkers (i.e., Zonulin, LBP, Claudin-3, I-FABP) assessed in a subset of these participants (n = 23) further evidenced blast exposure associates with IP. The observed symptom profile was consistent with mild traumatic brain injury and was further associated with changes in bacterial translocation and intestinal permeability, suggesting that IP may be linked to a decrease in cognitive functioning. These preliminary findings show for the first time within real-world military operational settings that exposures to blast can contribute to IP.

## 1. Introduction

Across all 9/11 Active Duty and Veteran service members, ~20% have reported at least one mild traumatic brain injury (mTBI), with more than one-third of these having been attributed to blast-related injuries [1]. In military settings, exposures to blast can be incurred from incoming, uncontrolled high-level blast exposures such as IEDs or rocket-propelled munitions, or repetitive, controlled low-level blast from outgoing user-directed munitions such as Carl Gustaf recoilless rifle fire and explosive breaching charges (for a review, see [2]). Investigations in experimental models of blast as well as human studies reveal that blast-related neurotrauma pathologically differs from blunt-force trauma (e.g., vehicular accidents or falls) [3,4,5], in that exposures to blast are associated with axonal shearing and diffuse axonal injury [2,6,7,8]. Although the entire body is susceptible to blast injury, the brain seems particularly vulnerable [9]. A blast wave not only reflects off the skull, but the resulting energy of the shock wave is also absorbed by the brain tissue [10]. Also, the kinetic injury from thoracoabdominal compression can result in the transmission of forces to the brain by blood vessels [11] and further potentially result in gut permeability. These exposures typically lead to sub-concussive blast effects that do not yield a medical diagnosis yet do result in constellations of symptoms, including headaches and diminished cognitive functioning, that are in line with symptoms typically reported by individuals experiencing mTBI [12,13,14].

Due to the scarcity of data on the effect of blast on the human gut microbiota, researchers have used mTBI as a proxy. For mTBI, researchers have observed intestinal dysfunction and systemic inflammation after an event. Patients with TBI present with abdominal pain, gastric distension, altered intestinal motility, constipation, and/or ulcers, with these individuals sustaining more complications and experiencing worse outcomes with a chronic dysfunction of the gastrointestinal system and disability [15]. TBI can also lead to increased gut/intestinal permeability (IP). Following TBI, IP is associated with a concomitant reduction in intestinal tight junction protein expression. Intestinal epithelial cells function as a physical barrier between the intestinal lumen and the underlying vasculature and lymphatics that is supported by the structural integrity of cellular tight junctions, which are composed of a series of interacting transmembrane proteins, including claudins, Occludin, and junctional adhesion molecules. Emerging findings from preclinical animal models of TBI have shown that the decreased expression of the intestinal tight junction proteins ZO-1 and Occludin are linked to the loss of tight junction barrier integrity and IP [16]. IP has been reported as early as 2 h following TBI and, as such, the disruption of the anatomic and functional integrity of the gut can result in bacterial translocation into the circulatory system. 

Several established IP biomarkers used in clinical settings include Zonulin (haptoglobin 2 precursor), Lipopolysaccharide-Binding Protein (LBP), Claudin-3, and intestinal Fatty Acid Biding Protein (I-FABP). Zonulin, an intestine-synthesized protein [17], serves as an IP marker by directly enhancing the permeability of intestinal tight junctions between cells of the digestive tract [18,19]. LBP is a soluble acute-phase protein synthesized by intestinal cells that can be used to monitor gut leakage by detecting the presence of lipopolysaccharide (LPS). LBP can recognize and bind to LPS [20,21], the main component of the outer membrane of gram-native bacteria [22]. After binding, LBP then presents LPS to cell surface pattern recognition receptors (PRRs) such as CD14 and TLR4, to elicit subsequent immune response to infection [23,24]. Claudin-3, also known as CLDN3, is a protein which is encoded by an intron-less gene *CLDN3* in humans. It is an integral membrane protein and a component of tight junction strands and, therefore, serves as a marker of paracellular barrier integrity loss [25]. I-FABP is a cytosolic protein found within mature enterocytes and is a marker of enterocyte turnover and intestinal cell wall damage. Moreover, investigations of the human blood microbiome in neurodegenerative and psychiatric disorders have also provided evidence of bacterial translocation and potential for IP. However, since blood has been generally considered a sterile environment that lacks microbes, studies on the human blood microbiome have received little recognition until recently with the availability of a large compilation of reference microbial genomes, where increased microbial community variability in blood samples of schizophrenia and Alzheimer’s disease patients have been reported [26,27].

The present study investigates microbial variability and alterations in IP biomarkers using these approaches in blood samples collected from military service members participating in a repeated blast training event, breaching exercises. “Breachers” and “breaching” refer to an occupational discipline in the military and law enforcement in which explosives can be used to make entry to secured structures, and breachers, during training for this discipline, are repeatedly exposed to blast shockwaves. Breachers, and others who fire heavy weapons, report mTBI-like symptomology during these training exposures. Within this cohort of breachers and corresponding blood sample specimens collected during the course of the breacher training, we aimed to identify transcriptional and protein alterations associated with blast-induced IP and associated symptom reporting. We hypothesized that exposure to blast will (1) induce increased bacterial diversity in blood; (2) induce alterations in the levels of gut permeability protein biomarkers in circulation including Zonulin, LBP, Claudin-3, and I-FABP; and (3) that these molecular alterations related to IP will be associated with acute blast-related exposure symptomology. 

## 2. Results

The participants’ average age was 30.0 ± 6.5 years, with the majority of participants (83.3%) serving as combat engineers (MOS 12B) as their military occupational specialty. Participants’ self-reported history of mTBI and number of career breaching events were recorded prior to engagement in training and are shown in Figure 1; a total of 18 individuals reported a prior history of mTBI; however, there was no association between history of mTBI and career breaching history (*p* = 0.614) ranging from 0 to 400+ self-reported career breaching events (see Table 1 for full reporting of demographic information).

Microbial variability in blood samples was collected from 30 military service members participating in breaching tactical and training exercises (Figure 1). Following removal of blood RNA sequencing reads that mapped to the human transcriptome, candidate microbial sequences were identified, and the relative prevalence of the top three microbial phyla found in blood throughout the breacher training were characterized (Figure 2a), showing no significant differences in the prevalence of these microbial phyla following exposures to blast (*p* = 0.5933, 0.8177, 0.7190 for Proteobacteria, Actinobacteria and Firmicutes, respectively). Microbial diversity (measured by alpha diversity) on the phylum level were assessed by performing ordered logistic regression analyses showing significant stepwise increases in alpha diversity following exposure to blast (*p* = 0.049, Figure 2b). For a subset of participants (n = 24), changes in IP biomarkers (Zonulin, LBP, Claudin-3, I-FABP) were assessed during training. Using linear mixed effect models, accounting for age as a covariate with random intercepts for each subject, significant differences in Zonulin, LBP, Claudin-3, and I-FABP levels were observed across the three timepoints (Figure 3) and pairwise post hoc analysis, i.e., post- vs. pre- and follow-up vs. pre-analyses showed significant elevation in the levels of these biomarkers following exposure to blast (Table 2).

We found that both pre-post and pre-follow-up change in IP biomarker levels showed correlations with cumulative exposure with data available for 21 participants (Appendix A). Exploratory investigations involving blast-related symptoms in association with measures of alpha-diversity and IP biomarkers were also performed. The examination of symptom data collected serially with blood collection sampling during the training showed an elevated reporting of symptoms (for those symptoms with more than 10 subjects reporting at post timepoints) related to headache, concentration, dizziness, and taking longer to think acutely following blast (Figure 4) that dissipated 16 h following. For each symptom, participants were assigned to either of two groups based on changes in their self-reported symptoms (i.e., increasing vs. non-increasing) by comparing the symptom scales reported at pre vs. post per subject. Then, the increasing vs. the non-increasing groups by symptom were compared for differences in alpha-diversity and levels of IP biomarkers (separately) across timepoints (specifically post-pre), with effect size measured by Cohen’s d (Figure 5, and Appendix A). 

## 3. Discussion 

This study presents data acquired from military Breachers that show evidence of bacterial translocation into circulation, corroborated by an additional line of preliminary data that shows alterations in IP protein biomarkers, 1-to-16 h following military occupational training where individuals were exposed to blast. These data also show elevated reporting of symptoms of headache, dizziness, concentration, and taking longer to think by participants acutely following blast that dissipated 16 h following. These constellations of symptoms are in line with symptoms typically reported by individuals experiencing mTBI [13,14], and degradations in cognitive performance amongst individuals exposed to blast in military operational breaching during training that include the specific training protocol investigated in the present study [28]. Symptoms were further shown to be associated with changes in bacterial translocation and IP as measured by alpha diversity and levels of IP biomarkers, suggesting that IP may be linked to decreased cognitive functioning. To our knowledge, this is the first study that shows that exposures to blast in a military operational setting contributes to bacterial translocation and intestinal permeability along with associated cognitive symptoms, establishing the role of the gut–brain axis in blast-related sequalae.

This study has several limitations. As an observational human study conducted within military operational settings, clinician-administered assessments were not performed since it was not feasible and would have interfered with the operational duty of the participants. Also, as a human study, the inclusion of a control group is challenging. Since the protocol requires multiple days of military training exercises, typically, there are no available service members with comparable military occupational specialties in handling explosives and heavy weapons that could serve as controls. Hence, this study was based on a longitudinal design with repeated measures, where each participant’s pre-exposure is their own control when comparing symptom and biomarker measures pre- vs. post- and follow-up exposures to blast. Further, there was a lack of representation of both sexes among the study participants. Although the training protocol was open to both sexes, there were no females in this training cohort. As an observational study, the data were collected where blast exposure occurred in existing operational training environments, and they presently represent the bias in predominantly male service members that participate in these training protocols. There were also no longer-term longitudinal data on both bacterial species and IP measures following blast exposures, which were not feasible given the scope of the parent protocol under which the symptom and biospecimen data were collected, as well as the participants’ duties while in military service. In terms of biomarker detection, it is possible that the experimental approaches were not sensitive enough to detect subtle changes in IP biomarkers that could be investigated in future studies using other approaches as digital ELISA (or single-molecule enzyme-linked immunosorbent assay). 

Military service members frequently operate in extreme environments that challenge their health, in terms of cognitive and physical functioning. There is a growing recognition of the role of microbiota dysbiosis and deleterious health outcomes related to military operational exposures (e.g., psychological stress, sleep deprivation, environmental extremes (high altitude, heat, and cold), noise, diet (reviewed in [29])), and, notably, exposures to blast supported by the present study. These exposures can induce central stress responses that lead to altered gastrointestinal and immune function that potentially elicit unfavorable changes in gut microbiota composition, function, and metabolic activity, resulting in dysbiosis that further compromises gastrointestinal function and mucosal barrier and facilitates the translocation of gut microbes into circulation. The gastrointestinal tract tightly controls antigen trafficking by providing a dynamic barrier through both the transcellular and paracellular pathways [30]. Intercellular tight junctions are the key structures regulating the paracellular trafficking of macromolecules. Zonulin is a known physiological modulator of intercellular tight junctions synthesized within intestinal and liver cells [17] that modulates the permeability of tight junctions between cells of the wall of the digestive tract. Zonulin has been used as an IP biomarker for autoimmune diseases such as inflammatory bowel disease (Crohn’s disease [31]), as well as multiple sclerosis [32] and Schizophrenia [33]. In the present study, the plasma level of Zonulin was significantly increased at 1 h and 16 h post blast exposure, indicating that blast may contribute to the impairment of the gut barrier in the paracellular pathway (Figure 3 and Table 2), which was further supported by elevated bacteria alpha diversity in the blood transcriptome (Figure 2b and Table 2). 

To additionally capture the potentially complex effect of blast on gut permeability, we assayed levels of other molecules produced by the intestinal epithelial cells—such as fatty acid-binding proteins [34] (FABPs), lipopolysaccharide binding protein (LBP), and Claudin-3. Fatty acid binding protein (FABP) is one of the intracellular proteins, with a low molecular weight of approximately 15 kDa [35], that plays important roles in the transport and metabolism of long-chain fatty acids [36,37]. The FABP family of proteins and specifically intestinal-FABP (iFABP) is found primarily in the enterocytes of the jejunum and in the colon and is rapidly released into the circulation following intestinal mucosal tissue injury. A significant elevation of circulating levels of plasma I-FABP was detected 16 h following exposures to blast (Figure 3 and Table 2). It is possible that the cumulating I-FABP levels in circulation are not acutely detectable with the experimental assays used, with no observed differences pre- vs. 1 h post-blast exposure (Table 2). Circulating Claudin-3 IP biomarker levels have also been used for the detection of intestinal tight junction damage, paracellular localization, and expression throughout the jejunum, ileum, and colon [38]. Knockout studies of Claudin genes result in loss of tight junction barrier function, with the dysregulation of Claudin-3 gene reported in IBD (both Crohn’s disease [39,40] and ulcerative colitis [41,42]) and celiac disease [43]. Although increased circulating Claudin-3 was reported to be indicative of impaired gut barrier, data from the present study showed decreased levels of circulating Claudin-3 levels acutely 1 h post blast that returned to baseline prior to blast exposure within 16 h (Figure 3 and Table 2). It is possible that as an IP biomarker, circulating Claudin-3 expression might be reflective of chronic inflammatory conditions, rather than acute injury associated with exposures to blast. LBP is synthesized in hepatocytes and released into the bloodstream after glycosylation [44], and it is commonly used as a biomarker for the lipopolysaccharide (LPS) response following gut leakage and infection [45,46]. LBP, thereby, has been shown to serve as a gut permeability biomarker for several inflammatory disorders including celiac disease [47], systemic inflammatory response syndrome [48], heart failure [49], and obesity [50]. As the primary binding protein for LPS in circulation, LBP enhances the proinflammatory response and clearance of LPS [21]. LBP levels decreased within 2 h following blast exposure (Figure 3 and Table 2), suggesting possible elevations of inflammation post blast in response to bacterial translocation due to a leaky gut and blast exposure. Indeed, the inflammatory response to blast exposure is well-documented [51,52,53].

Immunological as well as endocrine, metabolic, and neural pathways are critically involved in the bidirectional communication pathways between gut microbiota and CNS, referred to as the “microbiota–gut–brain” axis. Broadly, the microbiota–gut–brain axis has been implicated in the pathogenesis of neurological disorders (e.g., Alzheimer’s and Parkinson’s), neuropsychiatric disorders (e.g., PTSD and depression), autoimmune disease (e.g., multiple sclerosis), CNS injuries (e.g., stroke, TBI, and spinal cord injury) [54,55], and headaches [56,57,58,59], with some of these disorders having greater prevalence in our warfighters and Veterans that may be attributable to their military experience [29]. Notably, combat-deployed service members with concussive blast injury, followed longitudinally, showed reductions in fractional anisotropy through imaging studies, which is indicative of chronic brain injury [60]. This suggests that injuries from subconcussive blast exposures may contribute to evolving brain injury pathology that manifests in a range of clinical symptoms [60] including chronic headaches [61,62,63] and cognitive impairment [60,64,65]. These are also reflective of symptoms of headache and cognitive problems reported acutely by military service members exposed to blast in operational settings, and specifically by participants in the present study with similar exposures to blast, which are consistent with concussion symptomology [66,67,68]. The elevated reporting of symptoms by these participants was related to headache, dizziness, concentration, and taking longer to think acutely following blast that dissipated 12 h following and was in line with constellations of symptoms typically reported by individuals experiencing mTBI [13,14]. Further, data from the present study show that these symptoms are associated with possible bacterial translocation from the gut and intestinal permeability measured via changes in alpha-diversity and IP biomarkers, respectively (Figure 5), with translational impact linking the effect of blast on the gut–brain axis with associated clinical symptoms. 

In conclusion, findings from this novel course of investigation suggest a possible role of blast exposure in IP and the importance of the gut–brain axis in blast injury. These findings have a major long-term potential for therapeutic impact in the manner by which we target clinical symptoms associated with exposure to blast and long-term sequalae through the adoption of a multi-targeted neuroprotective approach. Instead of prioritizing treatments directed toward single molecules or symptoms, findings from this study support investigations of interventions that modify multiple targets explicitly by stabilizing the gut microbiota, mucosal barrier, and systemic inflammation. This can be achieved through the deployment of the body’s own immunomodulatory machinery that may help to inform a personalized-medicine approach by optimizing the type of probiotic strain, dosage, and timing of delivery of psychobiotic treatments [55]. Specifically, findings from this study may have a widespread clinical impact, leading to a paradigm shift in the manner by which the military can mitigate injuries related to blast exposures through the delivery of probiotics for the treatment of microbiota dysbiosis and IP prophylactically [69].

## 4. Methods

All participants consented to participate in the study, and the human use protocol for interaction with the participants was approved by the Institutional Review Board (IRB) of the Walter Reed Army Institute of Research (WRAIR, Silver Spring, MD, USA; FYSA: WRAIR #2304) and chains of command prior to data collection. The procedures were followed in accordance with the ethical standards of the IRB, Army Regulation 70–25 and the Helsinki Declaration. 

### 4.1. Participant Training and Sample Characteristics

The data reflected three timepoints collected at one military training site. The study was conducted during military breaching training between August 2016 and July 2018. The training involved heavy wall-breaching exercises using explosive charges with a Net Explosive Weight (NEW) of 10 lb; individuals encountered at least two breaching charges where they were positioned in a breaching stack formation at a minimum safe distance of 40 ft (12 m) from the blast source (details on training environment described previously [28]). Thirty-three male subjects participated in a routine training protocol using explosives in the demolition of a concrete wall where cumulative exposure was measured as pressure (pounds per square inch, psi) times time (milliseconds, ms), or psi*ms, for each participant in the field with wearable sensors (BlackBox Biometrics Blast Gauge^®^, Airboss Defense Group, Jessup, MD, USA); however, blood sample specimens were available only for 30 participants investigated in the present study. Any individual actively assigned and participating in training was eligible to participate. Specifically, eligibility criteria were as follows: (1) at least 18 years old; (2) able to give informed consent; and (3) active duty military personnel or civilian law enforcement personnel. Note that all participants met the military’s physical requirement standards and were fit for duty. If not fit for duty, they would not be present on site. Although the training protocol was open to the participation of both male and female service members, no females participated during the data collection of this study. Blood samples were collected serially pre-blast (morning 7:30 a.m. to 9:00 a.m.), post-blast (afternoon 4:30 p.m. to 5:30 p.m.), on the training day, and upon follow-up the next day (morning 7:30 a.m. to 9:00 a.m.) and stored in −80 °C for downstream experiments. On average, the post-blast time point corresponds to 1 h and the follow-up to 16 h following exposures to blast during training. 

During the course of training, participants completed a 32-item, paper-and-pencil health symptom inventory at pre-, post-, and follow-up blast exposure timepoints in conjunction with each blood draw. Items on the symptom survey were the same as those of the Rivermead Post Concussion Symptoms Questionnaire [70,71], with additional items included relevant to capture additional effects previously observed as reported in blast exposure environments [67]. These items are consistent with concussion symptomology present in current clinical and research findings [67,68]. Items on this survey are on a 5-point Likert scale (0 “not experienced at all”, 1 “no more of a problem than before training”, 2 “mild problem—present but don’t really notice and doesn’t concern me”, 3 “moderate problem—I can continue what I am doing but I notice the problem”, 4 “severe problem—constantly present, feels like it could affect my performance”). Additionally, prior to the training, participants completed a paper-and-pencil survey on operational and medical history and symptom including age, duration of service, military occupational specialty (MOS), mTBI history, and number of prior experienced breaches reported on a 7-point Likert scale: none (0), 1 to 9 (1), 10 to 39 (2), 40 to 99 (3), 100 to 199 (4), 200 to 399 (5), and 400 or more (6). Data on most recent breaches or blast exposure prior to the present training were also collected. Recent prior breaches were reported within time frames on a 6-point Likert scale: past week (1), past month (2), past 6 months (3), past year (4), more than 1 year (5), and never (6). Data on blast exposure were reported by participants in a narrative within an open text box in the intake form. 

### 4.2. Total RNA Sample/Library Preparation Sequencing 

Blood samples were collected in Paxgene blood RNA tubes (PreAnalytix) according to the manufacturer’s instructions and were stored at −80˚C. RNA was extracted with Paxgene Blood RNA Kit (PreAnalytix, Hombrechtikon, Switzerland) and globin mRNA removed with Globin Clear Human Globin mRNA Removal kit (Ambion, Austin, TX, USA). Total RNA sequencing libraries were prepared from RNA samples with RNA integrity numbers, RINs > 6.0 (measured via BioAnalyzer) using the Illumina Stranded Total RNA Library Prep Kit with Ribo-Zero Gold in accordance with the manufacturer’s instructions. Briefly, 290–500 ng of total RNA was used for ribosomal depletion and fragmented by divalent cations under elevated temperatures. The fragmented RNA underwent first strand synthesis using reverse transcriptase and random primers followed by second strand synthesis to generate cDNA. The cDNA fragments underwent end repair, adenylation, and ligation of Illumina sequencing adapters. The cDNA library was enriched using 11 cycles of PCR and purified. Final libraries were then evaluated using PicoGreen (Life Technologies, Carlsbad, CA, USA) and Fragment Analyzer (Advanced Analytics) and sequenced on an Illumina HiSeq2500 sequencer (v4 chemistry) using 2 × 125 bp read lengths.

### 4.3. RNA-Seq Data Preprocessing and Microbial Bioinformatics Analysis 

RNA sequencing reads were aligned to the human reference hg19 using STAR aligner (v.2.7.5b) [72]. Following alignment of RNA sequencing reads, read pairs that mapped to the human genome were eliminated. We then performed normalization by sub-sampling the remaining reads for each sample to 100,000 reads, and then used FASTX (v.0.0.13) and Prinseq (v.0.20.4) to filter out low-quality and low-complexity reads; that is, reads with at least 75% of their base pairs having quality lower than 30 (FASTX) [73] and reads with sequences of consecutive repetitive nucleotides (Prinseq) [74]. Next, the remaining reads were realigned to the reference human genome and transcriptome (Ensembl GRCh38 transcriptome and Ensembl hg38 build) using Megablast aligner (Blast v.2.9.0+) to filter out any remaining potentially human reads. After QC, there was an average of 41,770 reads per sample. These remaining reads were used as candidate microbial reads. Phylogenetic classification was then performed using Kraken2 [75] to assign the filtered candidate microbial reads to the microbial genes from 33 distinct taxa on the phylum level. 

For the analysis of microbial diversity, alpha diversity on the phylum level within each sample was determined using the inverse Simpson index as follows: $$\alpha =\frac{1}{\lambda }=\frac{1}{\sum_{i=1}^{R}p_{i}^{2}}$$
where *R* is richness (the total number of types of phyla in the sample) and *p_i_* is proportional abundance of ith phylum. This index simultaneously assesses both the richness (corresponding to the number of distinct taxa) and relative abundance of the microbial communities within each sample. In particular, it enables effective differentiation between the microbial communities shaped by the dominant taxa and the communities with many taxa with even abundances. 

### 4.4. Intestinal Permeability Biomarkers Assayed Via ELISA 

Several IP biomarkers including Zonulin, LBP (LPS binding protein), Claudin-3, and Intestinal-Fatty Acid Binding Protein were assayed by a number of specific ELISA kits with a high sensitivity and specificity to measure concentrations as low as ng/mL in human blood samples analogously to previous reports that included studies on military service members and Veterans [76,77,78]. The plasma levels of Zonulin, LBP, and Claudin-3 were quantified using ELISA kits from MyBiosorce (catalog # MBS706368, MBS2024051 and MBS2023694, respectively). It should be noted that, since commercial ELISA kits such as Immundiagnostik Kit (Bensheim, Germany) are found to recognize off-target molecules like properdin instead of Zonulin [79], in the present study, to avoid such inaccurate measurements, we chose Zonulin ELISA MBS706368 kit (MyBioSource, San Diego, CA, USA) as well as applied internal positive controls with a series of standards for the measurement of Zonulin levels. The plasma level of I-FABP was measured using the ELISA kit from R&D Systems (catalog # DFBP20). All procedures followed the manufacturer’s protocols. In brief, 100 uL of each sample (undiluted plasma samples for Zonulin and Claudin-3 Elisa plates, 5-fold diluted and 500-fold diluted plasma samples for I-FABP and LBP ELISA plates, respectively) and a series of standards were added to each well with 2 h incubation at room temperature (RT), followed by three washes to remove unbounded substances. Detection antibody was then added to each well and incubated for another 2 h at RT. Three-time wash was then repeated to remove extra detection antibody. A total of 100 uL of HRP conjugate was added to each well and incubated for 20 min at RT. To remove the unbounded HRP conjugate, the wells were washed three more times, followed by the addition of 100 uL of a substrate solution to each well and incubation for 20 min at RT to react with the HRP conjugate. Finally, a 50 uL or 100 uL stop solution was added to each well to stop the HRP reaction. 

The concentrations of target molecules were then calculated based on the optical density of each, well determined by a microplate reader set to 450 nm and 540 nm, and calculated with CurveExpert professional [80,81] to find the best-fit curve formula and corresponding concentrations based on the optical density of the samples, standards, and blank controls. All IP biomarkers were measured in duplicate, and the average concentrations were calculated for subsequent analyses.

### 4.5. Data and Statistical Analyses 

Statistical and data analyses were performed using R 4.3.0 [82]. For each of the top three most prevalent phyla, the *t*-test was used to investigate the pre- to post-exposure change in the prevalence and whether the change in prevalence was associated with prior history of mTBI or cumulative history of blast exposures. An ordered logistic regression model was used, with the alpha diversity of each sample as outcome and timepoint as an ordinal response, to test whether alpha diversity changed stepwise over time following exposure to blast. The average concentration of IP biomarkers was used in mixed effect models as repeated measures, with time as predictor, adjusting for age as covariate, and subject-specific random intercepts. To compare the timepoints pre- vs. post- and pre- vs. follow-up, the pairwise comparisons were run using the “emmeans” function in R. Significance levels were adjusted for multiple testing by applying a Benjamini–Hochberg adjustment [83]. In exploratory analyses, we calculated and tested the significance of the correlation of the IP biomarker change from pre- to post- and follow-up with the cumulative exposure measured in pressure times time (psi*ms). To investigate the association between elevated-reporting symptoms with measures of alpha-diversity and IP biomarkers, for each symptom, we assigned participants to either of two groups based on changes in their self-reported questionnaire (i.e., increasing vs. non-increasing by comparing the symptom scales reported at pre- vs. post- per subject). Then, the increasing vs. the non-increasing groups by symptom were compared for differences in alpha-diversity and levels of IP biomarkers (separately) across timepoints (specifically post-pre). For these comparisons, given the small sample sizes, we computed the effect sizes measured by Cohen’s d, rather than performing significance testing. 

## Figures and Tables

**Figure 1 ijms-25-03549-f001:**
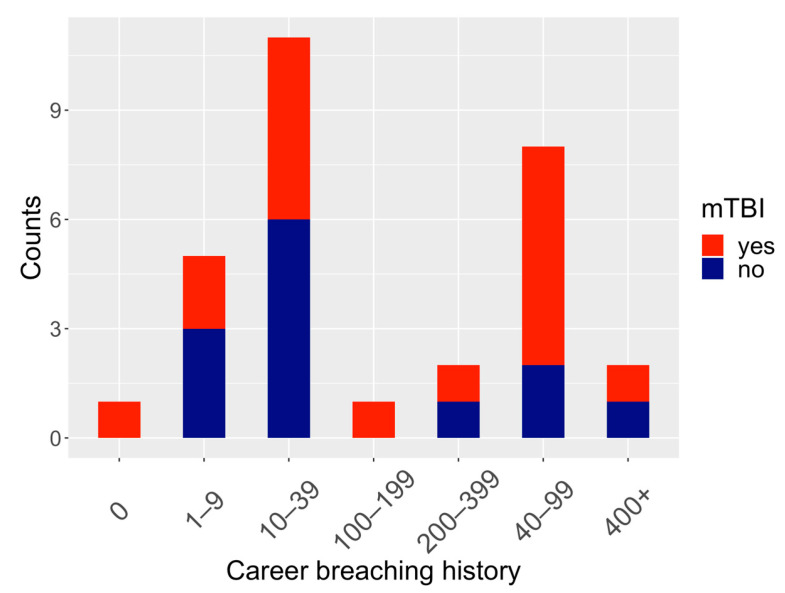
Number of self-reported career breaching experiences at baseline prior to start of training exercises including 30 participants with/out history of mTBI (denoted in red and blue, respectively).

**Figure 2 ijms-25-03549-f002:**
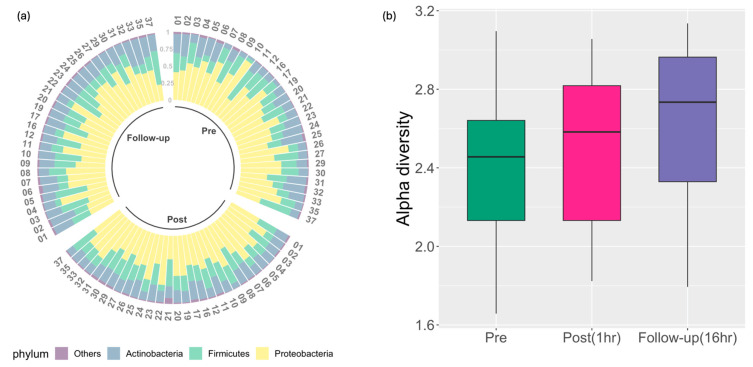
(**a**) Distribution of relative abundances of microbiota at phylum level at three time points, i.e., pre-blast, post-blast, and follow-up. Phylogenetic classification was performed to assign filtered candidate microbial reads to the microbial genes from 33 distinct taxa on the phylum level. Top three phyla, which include Proteobacteria, Firmicutes, and Actinobacteria, are shown in different colors depicting their proportions of RNA-seq reads mapped. From the vertex arranged clockwise are samples from 30 participants corresponding to pre-, post-training, and follow-up (1 h and 16 h post blast, respectively). Each participant is assigned a sequential ID number starting with ID number from 01 to 37. All “other” low abundant bacteria are grouped together. (**b**) Boxplot shows distribution of alpha diversity at phylum-level at each timepoint using time series blood RNA-seq data, i.e., pre-post and follow-up blast timepoints (i.e., 1 h and 16 h post blast, respectively). Significant (*p* = 0.049) stepwise increase in alpha diversity is observed following exposure to blast. Alpha diversity is derived using the inverse Simpson index, which simultaneously assesses both richness (corresponding to the number of distinct taxa) and relative abundance of the microbial communities within each sample.

**Figure 3 ijms-25-03549-f003:**
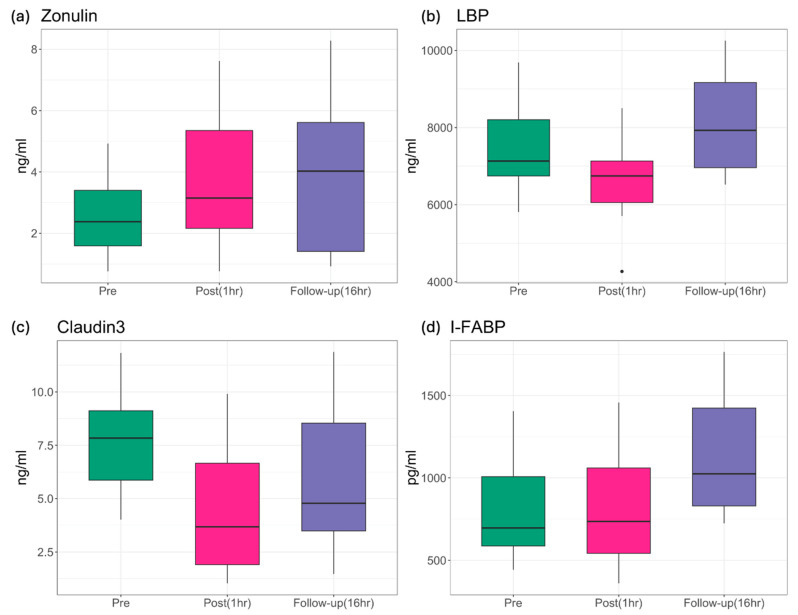
Levels of IP biomarkers in blood measured at each timepoint. Data are shown for (**a**) Zonulin, (**b**) LBP, (**c**) Claudin-3, and (**d**) I-FABP, respectively.

**Figure 4 ijms-25-03549-f004:**
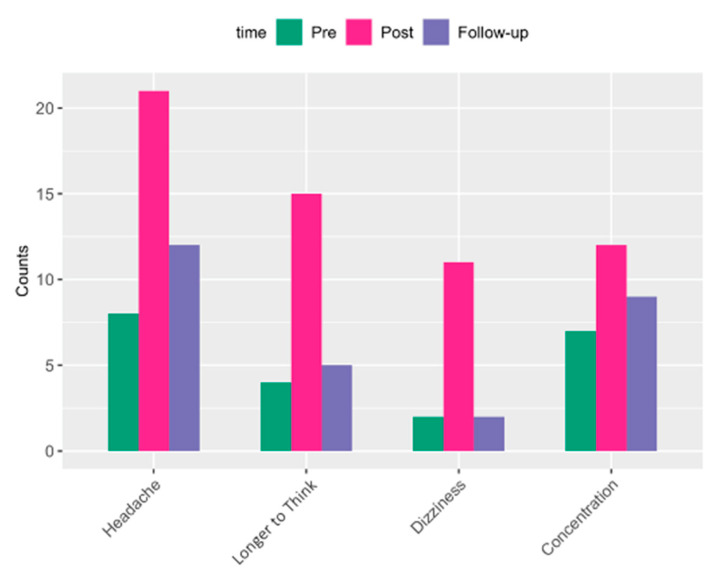
Symptom response frequencies from self-report surveys at each time point throughout the training (pre-blast, post-blast, and follow-up), showing elevated symptoms related to headache, slowed thinking, concentration, and dizziness, directly following exposure to blast (shown in pink bars).

**Figure 5 ijms-25-03549-f005:**
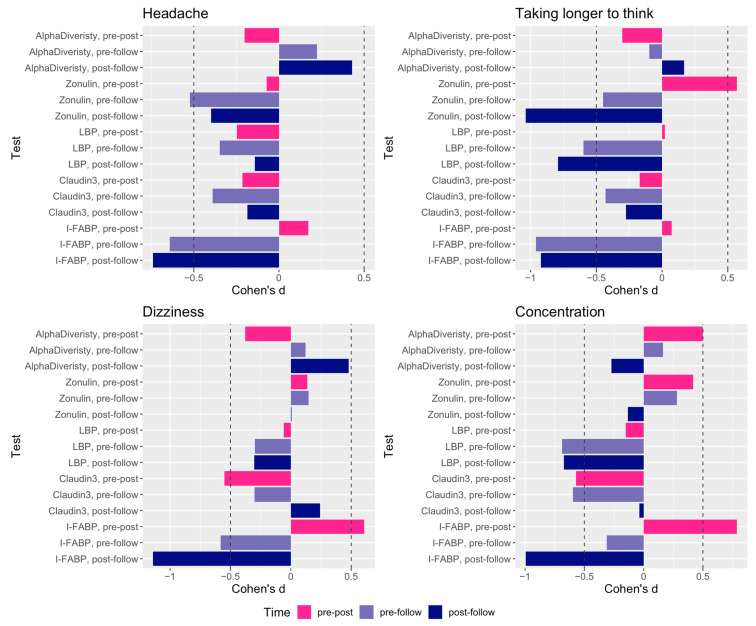
Association of changes in self-reported symptoms vs. changes in alpha-diversity and IP biomarkers, shown as effect sizes measured by Cohen’s d. For each elevated symptom self-reported, participants were assigned to either of two groups, i.e., increasing vs. non-increasing by comparing the symptom scales reported at pre vs. post per subject. Then, the increasing vs. the non-increasing groups by symptom were compared for differences in alpha-diversity and levels of IP biomarkers (separately) across timepoints (specifically post-pre). Note, negative value corresponds to lower levels of an IP biomarker or alpha diversity post-blast. Dashed lines are cutoffs between small and medium magnitude effect sizes.

**Table 1 ijms-25-03549-t001:** Summary of demographic information.

Age in Years, Mean (SD)	30 (6.25)
Duration of Service in years, mean (SD)	8.90 (6.25)
MOS, No. (%)	
12B/Combat Engineer	25 (83%)
31B/Military Police	3 (10%)
12H/Construction Engineer Supervisor	1 (3%)
19D/Calvary Scout 11B/Infantryman	1 (3%)
History of mTBI, No. (%)	
Yes	18 (60%)
No	12 (40%)
Career breaching history, No. (%)	
0	1 (3%)
1–9	5 (17%)
10–39	12 (40%)
40–99	7 (23%)
100–199	1 (3%)
200–399	2 (7%)
400+	2 (7%)
Time of last blast, No. (%)	
Past week	29 (97%)
More than a year ago	1 (3%)

**Table 2 ijms-25-03549-t002:** Linear mixed effect models, accounting for age as a covariate with random intercept for each subject, were used to test differences between timepoints in Zonulin, LBP, Claudin-3, and I-FABP levels. *p*-values of ANOVA and pairwise post hoc comparisons, i.e., post- vs. pre-and follow-up vs. pre-analyses are reported.

*p*-Value	Zonulin	LBP	Claudin-3	I-FABP
ANOVA	0.0018	0.0002	6.24 × 10^−5^	8.34 × 10^−5^
pre vs. post	0.0029	0.0222	3.8 × 10^−5^	0.7870
pre vs. follow-up	0.0029	0.0513	0.0115	0.0003
post vs. follow-up	0.9680	0.0001	0.0412	0.0003

## Data Availability

The datasets presented in this article are not readily available because such information as the site location, data collection date, date of birth, and gender, may reasonably allow for the identification of participants in the study. Deidentified raw data (not including these variables) supporting the conclusions of this article can be made available by the authors on the condition that institutional and ethical requirements for sharing the data are met.

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
