# Peer review of "Association of Blast Exposure in Military Breaching with Intestinal Permeability Blood Biomarkers Associated with Leaky Gut"

_ijms, 2024, doi:10.3390/ijms25063549_

Round 1

Reviewer 1 Report

Comments and Suggestions for Authors

Questions for Authors
A) What was the training environment? Was this a SFAUC course? 
B) When was this data collected? Was BETS collected? If not, why?
C) What was the basis for the collection timeline (1hr, 16hr)?
D) As serving as individual controls through pre vs post, what was the avg time from participants last previous blast exposure to “pre-blast” collection? I understand that it is nearly impossible to find naïve subjects, however, when examining low-level blast, this information is critical. Without knowledge of previous blast exposures, which including simple M4 firing (14” barrel, DoD standard, creates 4PSI in an open environment), the pre-collection data could be compromised. Please either include time since last low-level blast exposure or justify and include this as a limitation in the discussion.  
E) Was OSU-TBI used? If no, why not as it’s largely considered the standard to record previous TBI hx. 
F) What was the working definition of mTBI? Breachers using rigids or jelly rolls have a minimum safety distance (MSD) of ~6’ with a >4PSI. 

G) What was the standard explosive charge and what was the MSD during this training? 
H) During the explosives course, were flashbangs (concussive grenades) used, and if so was:
   i) the number of flashbangs recorded?

   ii) were the flashbangs single sequence, 3 sequence, or 6 sequence? 
I) In figure 6, why was post-blast vs follow up not statistically examined? 

Required changes for consideration for approval:
1)
 Intro- “with more than one-third of these having been attributed to blast related injuries from improvised explosive devices (IEDs) or rocket propelled grenades[1].”
- The authors intro only briefly introduces blast injuries, to which there is a growing body of literature on. 
The authors should: 
     i) expand the intro to provide context how blast injuries are different than blunt-force.
     ii) provide context to the reader on the differences (and importance) of various blast exposures. The intro currently highlights high-energy (incoming/uncontrolled) blast exposures, while the sample group (breachers) is experiencing repetitive low-level (outgoing/controlled) blast exposures. 

2) Demographics table for the group. Avg. age, ethnicity, time in service, number of TBIs, number of blast exposures, time since last TBI, time since last blast exposure, enlisted vs officer %, and ideally at least MOS series (11 series, 18 series, etc. )
-These factors have demonstrated to be strongly associated with number of TBIs, biomarker development, and blast exposures (see various papers from Jessica Gill). 
If the above cannot be produced, these must be included as strong limitations to the study that severely limit the interpretation of these data.

3) In Figure 6, please include post vs follow up. 

Comments on the Quality of English Language

Fine

Reviewer 2 Report

Comments and Suggestions for Authors

Dear Editor,

I read with great interest the manuscript draft titled “Association of Blast Exposure in Military Breaching with Intestinal Permeability Blood Biomarkers Associated with Leaky Gut” which has been submitted for publication in MS. 

The authors studied the effect of blast exposure on patients' symptomatology and a battery of interstitial permeability biomarkers, concluding that the symptoms could, at least in part, be attributed to changes in the gut-brain axis.

The study is interesting and novel and seems to comply with the international ethical requirement.

However, the authors need to address the following issues:

  1. The eligibility criteria should be set a the methodology section. 
  2. The explore (blast) should be explained in greater detail.
  3. Sample demographics should be transferred to the results section, and a summary table is required.
  4. Cohen’s d should be reported along with its 95% CI.
  5. The spaghetti plot in Figure 3 is redundant.
  6. Due to the small number of participants, the authors should present the before-after-follow-up measurements of the biomarkers under study in a table.

Otherwise, it is a well-written draft which is supported by the appropriate references. The conclusions are justifiable by the study findings and I anticipate that it is going to have a high impact in the field of military and civil trauma medicine. All the above changes could be performed without major changes in the manuscript draft. Therefore, I recommend minor revisions. 

Best regards.

Reviewer 3 Report

Comments and Suggestions for Authors

This study investigated intestinal permeability (IP) in 30 male
breachers exposed to controlled low-level explosive blast during
training. The findings suggest that exposure to blast can contribute
bacterial translocation into circulation, as evidenced by significant
stepwise increase in microbial diversity and levels of IP biomarkers
(i.e., Zonulin, LBP, Claudin-3, I-FABP). Furthermore, the observed
symptom profile was consistent with mTBI and was further associated with
changes in bacterial translocation, suggesting that IP may be linked to
a decrease in cognitive functioning.

This study is well designed and provides important new insights into the
pathogenesis of blast-induced alterations in military health and
performance. The manuscript is well written and referenced, however, the
introduction is long and could be shortened by reducing some
well-established background on the biophysical characteristics of
explosive blast waves and their interactions with the surrounding
environment. 

Author Response

We thank the reviewer for this feedback and have, in the revised manuscript shortened the introduction by eliminating the description of the biophysical characteristics of explosive blast, since this is commonly known as suggested by the reviewer (see revised paragraph 1 of the Introduction). 

Round 2

Reviewer 1 Report

Comments and Suggestions for Authors

Dear Authors, 

First, thank you for your reply. However, I would like to address your rebuttal.

Your demographics table does not satisfy critical components and metrics needed in order to interpret your results. Namely, blunt force TBI has been associated with GI and gut permeability issues. Not conducting a proper OSU-TBI ID is a severe deficit in design. While I understand that conducting a clinical instrument during a training exercise can be difficult, this does not excuse foregoing the capture of such critical information particularly on smaller cohort/sample size. Many others have done so with far more complicated populations (Special Operations). Furthermore, stating that it was not a deployable instrument due to environment and time. However, blood draws were performed presumably by clinical personnel, while the OSU-TBI-ID, as stated by the developing institution, takes 3-5 mins to complete. Unfortunately, the overlap between blast induced pathophysiological effects and blunt force TBI is currently too intertwined, and without context of the number of TBIs and the time since last TBI, the current results cannot be interpreted as to their relationship to the primary hypothesis. This data becomes even more imperative when participants are used as baseline for "pre-exposure". 

While the data is interesting, again, it cannot be interpreted. Because the study has concluded, Obtaining these data would be very difficult as the study took place nearly 5-6yrs ago and finding the participants is unlikely. 

For that reason, I must reject the paper. 

Thank you and good luck. 

Author Response

We thank the reviewer for these comments. As noted by the editor, these comments may be subjective by nature, and thus, no response is provided.